# Structured Docosahexaenoic Acid (DHA) Enhances Motility and Promotes the Antioxidant Capacity of Aged *C. elegans*

**DOI:** 10.3390/cells12151932

**Published:** 2023-07-26

**Authors:** Ignasi Mora, Alejandra Pérez-Santamaria, Julia Tortajada-Pérez, Rafael P. Vázquez-Manrique, Lluís Arola, Francesc Puiggròs

**Affiliations:** 1Brudy Technology S.L., 08006 Barcelona, Spain; 2Eurecat, Centre Tecnològic de Catalunya, Nutrition and Health Unit, 43204 Reus, Spain; alejandra.perez@estudiants.urv.cat; 3Laboratory of Molecular, Cellular and Genomic Biomedicine, Instituto de Investigación Sanitaria La Fe, 46026 Valencia, Spain; julia_tortajada@iislafe.es (J.T.-P.); rafael_vazquez@iislafe.es (R.P.V.-M.); 4Joint Unit for Rare Diseases IIS La Fe-CIPF, 46012 Valencia, Spain; 5Centro de Investigación Biomédica en Red de Enfermedades Raras (CIBERER), 28029 Madrid, Spain; 6Nutrigenomics Research Group, Departament de Bioquímica i Biotecnologia, Universitat Rovira i Virgili, 43007 Tarragona, Spain; lluis.arola@urv.cat; 7Eurecat, Centre Tecnològic de Catalunya, Biotechnology Area, 43204 Tarragona, Spain

**Keywords:** aging, *C. elegans*, healthspan, omega-3, PUFA, docosahexaenoic acid, physical decline, structured lipids, DAF-16/FOXO, oxidative stress, antioxidant, nematode, cognitive decline

## Abstract

The human lifespan has increased over the past century; however, healthspans have not kept up with this trend, especially cognitive health. Among nutrients for brain function maintenance, long-chain omega-3 polyunsaturated fatty acids (ω-3 LCPUFA): DHA (docosahexaenoic acid) and EPA (eicosapentaenoic acid) must be highlighted, particularly structured forms of EPA and DHA which were developed to improve bioavailability and bioactivity in comparison with conventional ω-3 supplements. This study aims to elucidate the effect of a structured triglyceride form of DHA (DHA-TG) on the healthspan of aged *C. elegans*. Using a thrashing assay, the nematodes were monitored at 4, 8, and 12 days of adulthood, and DHA-TG improved its motility at every age without affecting lifespan. In addition, the treatment promoted antioxidant capacity by enhancing the activity and expression of SOD (superoxide dismutase) in the nematodes. Lastly, as the effect of DHA-TG was lost in the DAF-16 mutant strain, it might be hypothesized that the effects of DHA need DAF-16/FOXO as an intermediary. In brief, DHA-TG exerted a healthspan-promoting effect resulting in both enhanced physical fitness and increased antioxidant defense in aged *C. elegans*. For the first time, an improvement in locomotive function in aged wild-type nematodes is described following DHA-TG treatment.

## 1. Introduction

Thanks to medical advancements and life science technology, individuals are now living longer than ever before, but healthspan has not kept up with this trend [1]. To improve the quality of life of the elderly, it is crucial to consider the role of preventive health interventions. As nutrition is closely associated with the prevention of age-related diseases, there is a growing demand for appropriate dietary patterns in which food supplements and functional foods play a role in extending the healthspan of aging people and preventing age-associated dysfunctions [2].

Among healthspan-promoting nutrients, long-chain omega-3 polyunsaturated fatty acids (ω-3 LCPUFA) must be highlighted, specifically docosahexaenoic acid (DHA) and eicosapentaenoic acid (EPA). Increased intake of ω-3 LCPUFA has been associated with better cognitive function, a lower risk of developing dementia [3,4], slower rates of muscle mass loss [5], and the improvement of severe neurodegenerative diseases [6]. These healthspan-promoting properties of the ω-3 LCPUFA are the consequence of its inflammatory-resolving and antioxidant properties which delay cellular senescence and help the immune system [7,8]. An increased presence of these fatty acids in the membrane phospholipids of cells causes, firstly, changes in the membrane fluidity and signaling platforms (lipid rafts), thus modifying signaling transduction [9]; secondly, the activation of transcription factors that promote cell defense [10], and thirdly, enhanced production of eicosanoids and SPMs (specialized pro-resolving mediators) which are powerful short-range hormones modulating inflammation, immune response, and reproductive process, and are derived from ω-3 LCPUFA. They are known as prostaglandins, leukotrienes, resolvins, and lipoxins and are synthesized by cyclooxygenase, lipoxygenase, or cytochrome P450 enzymes [11].

Despite ω-3 LCPUFA being very prone to oxidation, becoming oxidized lipids that might be harmful, they have the ability—particularly DHA—to promote the activity of antioxidant enzymes and to improve the counteraction of oxidative stress [12,13]. The presence of DHA in membrane phospholipids of cells slightly increases ROS (reactive oxygen species) production, which stimulates the antioxidant capacity [14]. This phenomenon makes cells more resilient against foreign oxidative stress through several pathways: regulating antioxidant transcription factors [10]; enhancing the activity of antioxidant enzymes like superoxide dismutase (SOD), catalase (CAT) [14,15], and glutathione reductase (GR) [16]; and increasing reduced glutathione levels (GSH) which decreases oxidative damage [17].

Currently, structured forms of ω-3 LCPUFA have emerged as promising ingredients with powerful benefits for human health. This group of esterified ω-3 LCPUFA comprises a wide range of large structures, like triglycerides (TG) or phospholipids [18]. According to the findings from clinical studies, these complex structures of EPA and DHA have proven greater intestinal absorbance and membrane incorporation in human tissues in comparison to simpler forms of ω-3 LCPUFA, like ethyl-ester (EE) [19,20]. In consequence, powerful anti-inflammatory and antioxidant effects after their supplementation have been documented [21,22]. Nevertheless, due to the novelty of these structures and their use in supplementation, their molecular mechanism of action still needs to be elucidated through in vitro and pre-clinical studies [18].

The nematode *C. elegans* is an experimental model frequently used to study aging due to its lifespan characteristics, cellular simplicity, genetic amenability, and homology to humans [23] and has also emerged as a great model organism for nutrient screening in healthspan-promoting interventions [24,25]. Both mammals and *C. elegans* use and synthesize LCPUFA and eicosanoids, and these are thought to be vital for some functions of the nematodes, including correct neurotransmission [26], pharyngeal pumping [27], and sperm guidance in reproduction [28]. However, the *C. elegans* genome does not encode orthologs for some receptors of LCPUFA’s derived molecules [29], and in contrast with mammals, nematodes are capable of synthesizing all PUFA from de novo, interconverting ω-3 and 6 fatty acids (they are not essential fatty acids in nematodes) [30]. Therefore, LCPUFA has a different role as a substrate in *C. elegans*, and for this reason, the worm acts as a versatile model for the study of alternative functions of LCPUFA and its integration in development or aging.

At present, there is still little research assessing the effect of ω-3 PUFA supplementation in *C. elegans,* and its health-promoting effect in the nematode is debated. On the one hand, ω-3 PUFA seems to improve lifespan [31], reduce oxidative stress [32], ameliorate sarcopenia [33], and promote motility [29] in *C. elegans*, but on the other hand, depending on the dosage, ω-3 PUFA have been described as harmful due to increased lipid peroxidation, reducing the lifespan of the nematodes [34,35].

In this research, the healthspan effect of an experimental structured DHA-TG, which theoretically should demonstrate high bioactivity, is assessed using aged *C. elegans*. As recent literature encourages testing the healthspan-promoting interventions with aged *C. elegans* [36], the effects of treatments were monitored during the early stages of senescence, which provided some innovative data about lipidic supplementation in aged nematodes. Specifically, the effect of DHA-TG on locomotive function is characterized because ω-3 PUFA are described as beneficial for this function in nematodes [33]. In addition, a cognitive decline prevention effect was assessed as DHA has been described as necessary for efficient neurotransmission and mechanosensation in *C. elegans* [26,29,37]. A treatment with glyceryl trioleate (Oleic-TG) was added to monitor the energy intake and lipidic surplus due to the supplementation of fatty acids.

Moreover, because LCPUFA may act as mediators of insulin signaling and the transcription factor DAF-16/FOXO (forkhead transcription factor) in the nematodes [38], and as DAF-16 is enhanced in the majority of healthspan-promoting interventions with *C. elegans*, the effect of DHA-TG on locomotive function and gene expression was studied in a DAF-16 deficient strain.

## 2. Materials and Methods

### 2.1. C. elegans Strains and Maintenance

The wild-type (WT) *C. elegans* strain N2 (var. Bristol), the mutant *C. elegans* strain CF1038 *daf-16*(*mu86*), as well as the Escherichia coli feeding strain OP50 were obtained from the Caenorhabditis Genetics Centre (CGC, University of Minnesota, Minneapolis, MN, USA). All nematodes were maintained on standard nematode growth medium (NGM) agar plates at 20 °C, seeded with OP50 according to Brenner [39]. Before all tests, synchronized *C. elegans* populations were obtained by dissolving young adults in a sodium hypochlorite solution [40,41]. The obtained eggs hatched in M9 buffer (42.26 mM Na_2_HPO_4_, 22.04 mM KH_2_PO_4_, 85.56 mM NaCl, and 0.87 mM MgSO_4_) overnight and were transferred to new NGM control and treatment plates the following day. To inhibit reproduction, pre-fertile young adult worms were transferred to new control and treatment plates containing 100 µM 5-fluoro-2-deoxyuridine (FUdR) [42].

### 2.2. DHA-TG and Oleic-TG Treatment: Triglyceride Nanoemulsions

An oil-in-water phase solution at 10% was prepared by mixing DHA-TG (Brudy Technology, Barcelona, Spain) or Oleic-TG (Sigma-Aldrich, St. Louis, MO, USA) in M9 buffer with 20% of Tween 80 (Sigma-Aldrich, St. Louis, MO, USA) as emulsifier [43]. The mix was vortexed for 5 min and then sonicated (Q700, Qsonica Sonicators, Newtown, CT, USA) to create nanoemulsions for 3 min (Amplitude = 60) in 30 s intervals to avoid warming according to Colmenares et al. [44]. The atmosphere of tubes was replaced with nitrogen gas to avoid lipid oxidation.

The nanoemulsion stock solutions were diluted to the desired concentration using M9 buffer and Tween 80 to standardize all the treatments with a 0.009% Tween 80 concentration. A vehicle treatment solution with Tween 80 was also prepared following all the previous steps with a final concentration of <0.01% in M9. To feed the worms, the triglyceride nanoemulsions were dispersed over the surface of individual agar plates containing a lawn of *E. coli* OP50, as described by Colmenares et al. [44].

### 2.3. Nile Red Staining

Nile red (9-diethylamino-5-benzo[a]phenoxazinone) (Sigma-Aldrich, St. Louis, MO, USA) was dissolved in methanol, diluted in water, and homogenized for 24 h until complete dissolution. The dye solution was mixed with DHA-TG oil phase at a concentration of 1 mg/mL, and then the solvent (methanol) was removed by evaporation under nitrogen flux [43,44]. Nanoemulsions with stained oil were prepared and diluted as described above.

After 72 h of treatment, young adult worms were mounted onto 2% agar pads and anesthetized with a drop of 0.05 M sodium azide. Images were collected using a DM2500 (Leica, Wetzlar, Germany) vertical fluorescence microscope for scoring.

### 2.4. Lifespan Analysis

Lifespan assays were performed at 20 °C as previously described by Lionaki and Tavernakis [40]. Synchronized animals were transferred to NGM plates containing M9 1x (basal), Vehicle, Oleic-TG 50 µM, or DHA-TG 50 µM when indicated. Adults were scored manually as dead or alive every 1–2 days, and more than 60 animals were analyzed per condition. Nematodes that ceased pharyngeal pumping and had no response to gentle stimulation were recorded as dead. Worms that were male, crawled off the plate, or experienced non-natural death were censored. In some cases, FUdR was added to pre-fertile young adult worms to prevent the development of progeny.

### 2.5. Motility Assay

Motility was evaluated in worms using a thrashing assay, a measure of physical fitness that correlates with healthspan in *C. elegans* [45,46]. To assess the locomotive capacity of worms, the number of head thrashes was counted for 30 s per animal, and the average data were extrapolated to represent thrashes per min. Before scoring, each animal was acclimated in M9 medium for 30 s. We analyzed blinded at least 30 animals per condition. Each experiment was performed three times to obtain reproducible values.

### 2.6. Chemotaxis Assay

NGM agar was poured into 90 mm Petri plates and separated in quarters by labeling on the backside, as explained by Margie et al. [47]. The assay was performed on the 4th and 8th day of adulthood. FUdR was added to pre-fertile young adult worms to prevent the development of progeny. Approximately 500 treated or untreated individuals were collected in 15 mL tubes and washed at least 3 times with M9 buffer. At last, enough M9 volume to have a concentration of 5 nematodes/µL was poured into the tubes.

Then, 10 µL of control (C) or attractant (T) substance mixed with 1 M NaN₃ (Sigma-Aldrich, St. Louis, MO, USA) at 1:1 proportion was added to each quadrant to anesthetize arriving animals. Ethanol at 1% was used as C, and 2,3-Butanedione (Diacetyl, Sigma-Aldrich, St. Louis, MO, USA) at 1% was used as T. As soon as C and T were air-dried, a drop of 20 µL of worms was pipetted in the center of the assay plates [47].

After incubation for 1 h and 1.5 h (on the 4th and 8th day of adulthood, respectively) in the dark, the plates were directly moved to 4 °C. The different handling of the two aging groups is reasonable, given the dissimilar movement speed of the animals [48]. Thereafter, all nematodes in the four quadrants were counted to calculate the chemotaxis index (CI) = (n[T-quadrants] − n [C-quadrants])/n [T-quadrants + control-quadrants]. Control plates with just C quadrants were also included in the assay to discard possible nematode distributions linked to external inputs or genetic tendencies.

### 2.7. Preparation of Worm Lysates

Adult worms at 4, 8, and 12 days of adulthood (approximately 1000 adults) were harvested and washed at least three times with M9 buffer to remove all bacteria. After removing as much as possible of M9 buffer, worms were resuspended in 200–300 µL of lysis buffer: a mix of RIPA buffer (Sigma-Aldrich, St. Louis, MO, USA) and proteinase inhibitors cocktail (Thermo Scientific, Waltham, MA, USA), except for glutathione quantification where 5% sulfosalicylic acid (Sigma-Aldrich, St. Louis, MO, USA) was used. Worms were frozen in liquid nitrogen and thawed at 37 °C 5–6 times and vortexed between cycles. Then, worms were sonicated (Q700, Qsonica Sonicators, Newtown, CT, USA) in three cycles of 10 s (Amplitude = 60) and centrifuged at 12,000 rpm at 4˚C for 15 min. The supernatant was collected and stored at −80˚C. An aliquot was used for protein quantification by BCA (Bicinchoninic acid assay) (Thermo Scientific, Waltham, MA, USA) [48].

### 2.8. Lipid Peroxidation

Lipid peroxidation was quantified by assaying malondialdehyde (MDA) of worm lysates, which was determined using the thiobarbituric acid (TBA) method [49]. The thiobarbituric acid reactive substrate (TBARS) kit was performed following the manufacturer’s protocol (Cayman Chemical, Ann Arbor, MI, USA), and the absorbance was read at 532 nm with a SPECTROstar nano (BMG Labtech, Ortenberg, Germany). The values of MDA were calculated from the standard curve, and values were corrected for milligrams (mg) of protein in each sample.

### 2.9. Superoxide Dismutase Activity

Superoxide Dismutase (SOD) activity of worm lysates was measured using the SOD Colorimetric Activity Kit (Invitrogen, Waltham, MA, USA) according to the manufacturer’s instructions. The assay measures all types of SOD activity, including Cu/Zn, Mn, and FeSOD types, through the effect of the xanthine oxidase, which generates superoxide in the presence of oxygen. The colored product was read at 450 nm with a SPECTROstar nano (BMG Labtech, Ortenberg, Germany). The values of SOD were calculated from the standard curve, and data were corrected for mg of protein of each sample.

### 2.10. Glutathione Quantification

The glutathione content of the samples was assayed using a kinetic assay (Sigma-Aldrich, St. Louis, MO, USA) in which catalytic amounts of glutathione cause a continuous reduction of 5,5′-dithiobis-(2-nitrobenzoic) acid (DTNB) to TNB. The oxidized glutathione formed is recycled by glutathione reductase and NADPH, so total reduced glutathione (GSH) is quantified. The product, TNB, is assayed calorimetrically at 412 nm (SPECTROstar nano, BMG Labtech). The protocol was performed according to the manufacturer’s instructions, and the values of glutathione were calculated from the standard curve and were corrected for mg of protein of each sample.

### 2.11. RNA Extraction, Retrotranscription, and (RT)-qPCR

Treated worms at 4, 8, and 12 days of adulthood (approximately 1000 adults) were harvested and lysed as described above, but worms were resuspended in 500–1000 µL of TRIzol™ (Invitrogen) [48]. Total RNA from worms was isolated using the RNeasy Mini kit (Qiagen, Hilden, Germany) according to the manufacturer’s instructions. Before freezing at −80 °C, RNA concentration was determined with a Nanophotometer (Implen, Schatzbogen, Germany). Afterward, cDNA was synthesized using a High-Capacity cDNA Reverse Transcription Kit (Applied Biosystems, Waltham, MA, USA).

Using 10ng of cDNA as a template, qPCR analysis was performed on LightCycler^®^ 480 II (Roche, Basel, Switzerland) with LightCycler 480 SYBR Green I Master (Roche, Basel, Switzerland) following the manufacturer’s instructions. The primers used are listed in Appendix A. Each qPCR reaction was performed using more than three biological replicates in triplicate, and qPCR levels were normalized to the expression of *act-1*, which is predicted to be a structural constituent of the cytoskeleton. The fold change was normalized to that observed in vehicle *C. elegans* samples.

### 2.12. Statistical Analysis

Statistical analysis was performed using GraphPad Prism version 8 for Windows (San Diego, CA, USA). The results were plotted as the mean ± SEM (standard error of the mean) of at least two individual experiments and three or more biological replicates. Data were subjected to the Kolmogorov–Smirnov test for normality. For data with a normal distribution, Student’s *t*-test was used to compare pairs of groups, whereas two-way ANOVA followed by Tukey’s or Dunnett’s post-test was used to compare three or more groups with two different independent variables. All survival curves were analyzed with the Log-rank test using OASIS 2 (Online Application for Survival Analysis, Postech) [50]. The statistical significance was determined as *p* < 0.05.

## 3. Results

### 3.1. Internalization of Triglyceride Nanoemulsions

To assess if nematodes ingested the oil-in-water nanoemulsion with the method used, a fluorescence microscopy study was carried out after staining DHA-TG oil with Nile Red. Theoretically, nematodes could ingest a range of particle diameters between 40–500 nm [44]. Consequently, the red nanoparticles shown in Figure 1 might vary among those diameters.

The efficient internalization of nanoemulsions of DHA-TG is demonstrated in Figure 1 as judged by the presence of small red spheres in the pharynx and the intestinal region of the nematodes [44]. As described by Colmenares et al., images are organized to distinguish how fluorescence levels decrease as the concentration of DHA is diminished. Identical pictures with both red fluorescence and light can be consulted in the Appendix A to compare the size and synchronization of nematodes.

At the highest concentration of DHA (Figure 1A), there is the presence of Nile Red among all body parts. In contrast, at 500 µM of DHA (Figure 1B), the fluorescence of the nematodes is reduced. The lowest concentration (50 µM of DHA-TG) produces very low fluorescence, as represented in Figure 1C, where green arrows point to the nanoparticles present in the nematode’s pharynx.

Concentrations of 5 and 0.5 mM of DHA were only used to take pictures since, according to the literature, high doses of LCPUFA promote fat accumulation, lipid peroxidation, and damage to nematodes [35]. The concentration of 50 µM of DHA was chosen for the treatment with DHA-TG, as studies using ω-3 PUFA triglycerides supplementing *C. elegans* with our method used similar concentrations [32]. More pictures of DHA-TG nanoemulsions internalization can be consulted in Appendix A.

### 3.2. DHA and Oleic-TG Effect on the Lifespan of Wild-Type Nematodes

Although the main focus of the treatments is to prove an effect on the healthspan, which is not strictly associated with lifespan [45], the lifespan was controlled to elucidate whether the effect of the treatments promoted or decreased survival. In Figure 2, both Oleic-TG and DHA-TG show a trend toward extended lifespan, but not significantly (Table 1).

The treatment with DHA-TG under FUdR conditions shows a tendency (*p*-value of almost 0.05), as expressed in Table 1. The use of FUdR has been shown to extend lifespan, hypothetically, by promoting an adaptative response of the nematodes to stress induced by FUdR [51,52]. The effect of the treatments was studied with and without FUdR because, in some cases, opposite effects have been described in studies supplementing *C. elegans* when FUdR was used [51].

### 3.3. Effect of DHA-TG on the Behavior of Wild-Type Nematodes

#### 3.3.1. DHA-TG Improves Locomotive Function

To determine the locomotive function, the thrashing assay was used. This assay consists of counting how many “thrashes”, or body bends, the animals perform in liquid to avoid being pulled down by gravity [46]. The locomotive function is closely related to normal neurotransmitter release and healthy neuromuscular junctions, suggesting that improvement in motility may be related to reinforced neuromuscular connection [29]. Moreover, it has been shown that more body bend frequency is associated with a longer lifespan and overall healthspan [53].

Hypothetically, DHA-TG treatment should improve locomotive function, as ω-3 PUFA rescued motility in *C. elegans* mutant strains with deficient neuromuscular transmission [29,33]. Also, oleic acid is expected to be beneficial for motility, as it has been described in locomotive defective nematode strains [54].

In our study, DHA-TG treatment improved locomotive function in 12-day-old WT *C. elegans* independently of the use of FUdR (Figure 3), thus demonstrating the potential of DHA-TG to improve the physical fitness of aged worms.

The addition of FUdR (Figure 3B) decreased body bend frequency and emphasized the motility improvement after DHA-TG treatment in all the ages assessed. Pioneer studies testing FUdR already reported sluggishness of *C. elegans* after treatment with this substance [42], but it has not been documented as a motility reducer in thrashing assays. As FUdR affects reproductive function and causes stress in worms, it might be hypothesized that these effects are potential causes of slowness in the nematodes.

No differences between basal and vehicle groups were seen in lifespan or motility, and consequently, the basal group was discarded for the rest of the study, and the vehicle was used as the control group.

#### 3.3.2. Sensorial Response after DHA-TG Treatment

The experimental basis of the chemotaxis assay is to place nematodes in an area and observe the movement evoked in response to an odorant. A Petri dish is divided into four quadrants, two opposite quadrants marked as “Test” with the odorant, and two are designated as “Control” with ethanol. Anesthetic is placed in all test and control sites [47].

Nematodes in the four quadrants were counted to calculate the chemotaxis index (CI) with values from −1.0 to +1.0. The CI indicates the fraction of worms able to transform specific sensory stimuli into motor responses. Hence, better sensorial behavior is displayed when the CI is close to +1.0. As the worms advance in age, CI declines [48].

Wild-type nematodes at a young stage of adulthood show a CI between +0.7 and +0.9, as represented in Figure 4 [47]. Only nematodes on the 4th and 8th day were used because older *C. elegans* have reduced motility due to aging, which may affect the results [48]. LCPUFA are required for normal neurotransmission in *C. elegans* [26,29], and supplementation with PUFA phospholipids has been shown to increase the mechanosensory capacity of nematodes [37]. However, in our case, any difference was observed in CI values between treatments; therefore, the supplementation with DHA-TG did not produce changes in sensorial response.

### 3.4. Antioxidant-Promoting Effect of DHA-TG in Wild-Type Nematodes

DHA has been reported to improve oxidative stress counteraction in cell cultures and humans [18]; therefore, we wanted to assess the effect of DHA-TG on the antioxidant metabolism of nematodes. As better antioxidant capacity contributes to improved healthspan, according to the literature [39], a pro-antioxidant effect was expected after DHA-TG treatment.

Firstly, the assessment of lipid peroxidation (Figure 5A) showed no significant reduction in MDA equivalents with any treatment. It is suggested that MDA levels should increase with aging as ROS increases with senescence [55,56]; however, in our case, MDA levels decreased in 12-day-old nematodes. Few comparisons can be made since there are not many precedents quantifying MDA levels with age in wild-type *C. elegans*, but this phenomenon might be related to the reduction in the metabolic rate in aged nematodes that lowers mitochondrial function and consequent mitochondrial oxidative stress [57]. Furthermore, Shmookler Reis et al. describe that long-lived *C. elegans* strains have lower levels of lipid peroxidation [58], suggesting that the reduction in lipid peroxides may be a natural adaptation in favor of survival.

Secondly, glutathione concentration was monitored, as DHA is supposed to improve GSH synthesis [14]. The method quantified total glutathione since oxidized glutathione (GSSG) is converted to GSH by glutathione reductase (GR), but the ratio GSH:GSSG is expected to be 70:1 in young *C. elegans* adults [59]; hence, the results in Figure 5B represent mostly GSH levels. Treatments do not significantly enhance GSH synthesis compared to vehicle, but interestingly, the highest levels of glutathione were seen at 12 days of adulthood, which correlates with the lowest lipid peroxidation levels. A decrease in GSH and an increase in GSSG is expected as nematodes get old; however, irregular concentrations in aged nematodes have been reported [55]. There are not many precedents assessing GSH levels in aged *C. elegans*; for this reason, it is difficult to make a fair comparison.

Lastly, SOD was quantified, as increased SOD activity is related to healthspan improvement; SOD-deficient nematodes show reduced healthspan [60], and higher SOD levels have been related to improved lifespan [61]. DHA-TG treatment significantly improved SOD activity in comparison with the vehicle (Figure 5C), particularly in 4- and 12-day-old nematodes, and SOD activity increased as nematodes aged, as described [62].

### 3.5. Upregulation of Antioxidant Genes in Wild-Type Nematodes after DHA-TG Treatment

The next step was to determine the gene expression through quantifying mRNA levels of some key genes related to healthspan and lifespan in *C. elegans*. In addition, as DHA-TG treatment enhanced the SOD activity, the antioxidant response was further investigated.

Among the genes assessed Appendix A, just *sod-3*, *gst-7*, and *daf-16* showed significant differences against vehicle for each condition, as represented in Figure 6. The levels of *sod-3* are elevated on the 4th and 12th day of adulthood, which correlates with SOD activity (Figure 5C). Expression of *gst-7* and *daf-16* is only increased on the 12th day of adulthood, emphasizing the hypothetical healthspan benefits of DHA-TG in older stages of nematodes.

No significant differences were found in gene expression on the 8th day of adulthood (Figure 6) due to higher variability of replicates (larger SEM in comparison with the 4th and 12th). This age coincides with the end of the reproductive period of the nematodes, where morphological and molecular changes are present and affect gene expression. The reproductive decline is mainly driven by the Insulin/IGF-1 Signaling (IIS) Pathway (involves DAF-16/FOXO), which is the most prominent pathway in *C. elegans* aging research and connects the control of growth, reproduction, metabolism, and aging to nutrient status [63].

### 3.6. Loss of Effect in DAF-16 Deficient Nematodes after DHA-TG Treatment

The proper regulation of DAF-16/FOXO is detrimental to many health-promoting interventions. DAF-16 affects neuromuscular connection in *C. elegans* since it is involved in the expression of motility-promoting genes [63] and downregulates nematode motility when it is defective in neurons [64]. The mutant strain CF1038 (deficient for DAF-16) shows a significant decrease in motility in comparison with WT nematodes (N2) [65] which is confirmed in this study as shown in Figure 7A.

The motility-promoting effect in N2 observed in Figure 3B is lost when nematodes are deficient in DAF-16 (Figure 7B). Therefore, the data suggest that the mechanism by which DHA-TG can improve motility might involve DAF-16 and its downstream response.

In addition to motility evaluation, mRNA expression of the CF1038 strain after treatment with DHA-TG was quantified. As reflected in Table 2, significant differences were observed in *sod-*3 and *gst-7* in contrast with the results in WT (Figure 6). The gene expression of CF1038 was compared with N2 (WT) nematodes, and huge differences were observed in the *sod-3* expression, especially for 12-day adults. In the case of *gst-7*, there were also differences but not as high as *sod-3*. The expression of *sod-3* is very dependent on DAF-16, as described in the literature [66]; however, the diminishing expression of *sod-3* in the CF1038 strain among the elderly has not been characterized previously. These results support the idea that the healthspan-promoting effects of DHA-TG might be influenced by its antioxidant properties, and in the case of *C. elegans*, the activity and expression of the SOD enzymes family would be very involved.

## 4. Discussion

The treatment with DHA-TG in aged *C. elegans* improved their physical fitness and antioxidant capacity; consequently, the treatment exerted a healthspan-promoting effect. In nematodes, motility-promoting effects had already been characterized with ω-3 PUFA but only in motility-deficient strains [29,33]. For the first time, an improvement in locomotive function has been described in aged WT nematodes after ω-3 LCPUFA treatment.

The treatment with DHA-TG specially enhanced the activity and expression of the SOD enzyme family. Since the effect of DHA-TG is lost in DAF-16 deficient strain, and the expression of *sod-3* is very dependent on the factor DAF-16/FOXO [67], it might be hypothesized that healthspan-promoting effects of DHA-TG are closely related to the antioxidant response using DAF-16 as an intermediary, as stated in similar research [68,69].

According to references, antioxidant-promoting effects were already described in *C. elegans* after ω-3 PUFA use. Qi et al. characterized that α-linolenic acid (ALA) treatment activated the NHR- 49/PPARa and SKN-1/Nrf2 transcription factors, which are essential for the transcription of antioxidant enzymes genes. This effect was attributed to two main mechanisms: firstly, a phenomenon of mitohormesis which activated the mitochondrial stress response by a slight increase in ROS and subsequent adaptation due to the consumption of ALA, and secondly, an increase in oxylipins in nematodes through the oxidation of ALA that promoted the release of SKN-1 (Nrf2 ortholog) in the cytoplasm by reducing its inactivation [31]. Moreover, another recent study assessing linolenic acid (not specified as α) supplementation in *C. elegans* postulated that the reduction in ROS accumulation was promoted by mitochondrial repairment function via mitophagy. In that case, the antioxidant effect was explained by the activation of the DAF-16/FOXO transcription factor [33], as in our case, providing robustness to our hypothesis.

Although antioxidant effects with ω-3 PUFA are only described with linolenic acid, common parallelisms can be found with our results since we have also found DAF-16 as an intermediary of the effect of DHA [33]. ω-3 LCPUFA are very prone to oxidation, and it can be postulated that DHA supplementation might increase fatty acid oxidation, as happens with ALA treatment, which activates SKN-1 [31]. However, the effect of DHA-TG was accompanied neither by a significant reduction nor an increase in lipid peroxides (Figure 5A). In the nervous system of mammals, DHA is perhaps one of the most abundant fatty acids, but it does not cause lipid peroxidation and even shows antioxidant effects in nerve cells. Hypothetically, DHA acts as an indirect antioxidant in the nerve cell because it causes antioxidant effects through modulation of antioxidant gene expression and related pathways instead of directly counteracting ROS [69]. In our study, DHA causes a significant increment of *gst-7* and *sod-3* genes expression, probably, through DAF-16 intermediation. This indirect antioxidant capacity of DHA is very well documented in cell culture [14,70,71], but the exact mechanism is still unsolved. The main theory currently is that a phenomenon of hormesis reinforces the resistance to oxidative stress through adaptation to oxidized derivatives of DHA [18,72].

As far as is known, the mechanisms by which DHA improves locomotive function in motility deficient strains is through improved neurotransmission since LCPUFA are needed for the normal function of synaptic vesicles in *C. elegans* [29]. However, recent data emphasizes autophagy and counteraction of oxidative stress as the main benefits after ω-3 PUFA supplementation [31,33], which also has a high correlation with cognitive decline prevention [7]. Nevertheless, the probable mechanism by which DHA improves the healthspan of *C. elegans* may be a combination of all the effects mentioned.

In this sense, some ideas can be taken from the ω-6 PUFA literature since nematodes can interconvert ω-3 and ω-6 PUFA and may have common mechanisms of action [30]. For example, Chen et al. published a recent paper in which linoleic acid (LA, ω-6 PUFA) was supplemented to *C. elegans* and an interconnection between neurobehavior and antioxidant metabolism was described: high-doses of LA enhanced the activity of serotonergic neurons through activation of DAF-16 stress-related genes [73]. However, the effects of LA at high-dose were not reported as beneficial because the lifespan of the nematodes was reduced, whereas low doses of LA showed potential health-promoting effects [73]. The mechanism postulated by Chen et al. might fit as an explanation to the results shown in this article, but as ω-3 PUFA have distinct substrates and receptors in nematodes in comparison with ω-6 PUFA [30,34] and their study is focused on neurodevelopment stages instead of elderly, different effects can be expected and this hypothesis has to be proved in future research.

## 5. Conclusions

To summarize, muscular frailty and locomotive dysfunction are common characteristics in the elderly that significantly reduce the quality of life and increase the risk of injury in the aged population. ω-3 LCPUFA are promising bioactive compounds with preventive effects in age-related dysfunctions, and this study has proven that DHA-TG could potentially improve locomotive function in aged nematodes under lifetime supplementation. Moreover, our data reinforce the idea that the enhancement of antioxidant defense could help to counteract senescence dysfunctions as locomotive function decline and demonstrates *C. elegans* as a great model for first-stage studies about the bioactivity of structured lipids. Nevertheless, further studies are required to assess the potential correlation between the neuromuscular effects of DHA in the nematode and motility improvements.

## Figures and Tables

**Figure 1 cells-12-01932-f001:**
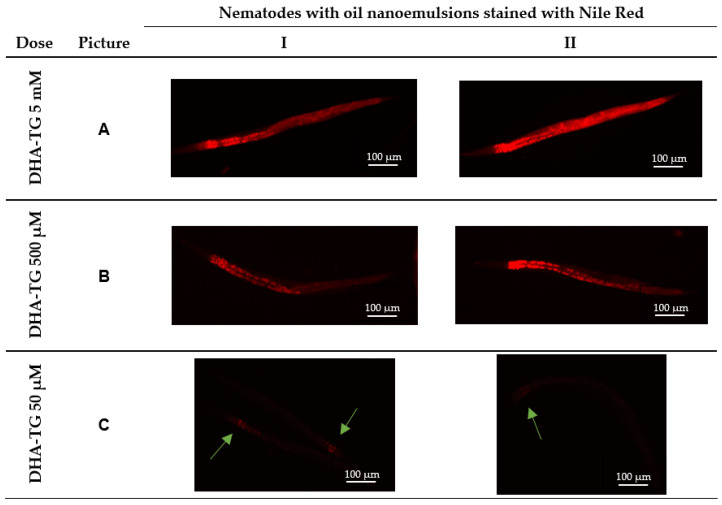
Confocal images of worms grown on agar containing nanoemulsions of DHA-TG oil stained with Nile Red. Images (**A**–**C**) show decreasing concentrations of DHA. Pictures I and II show different nematodes treated with the same dosage of DHA. Green arrows in the dark pictures C are pointing to the nematode’s fluorescence in the pharynx.

**Figure 2 cells-12-01932-f002:**
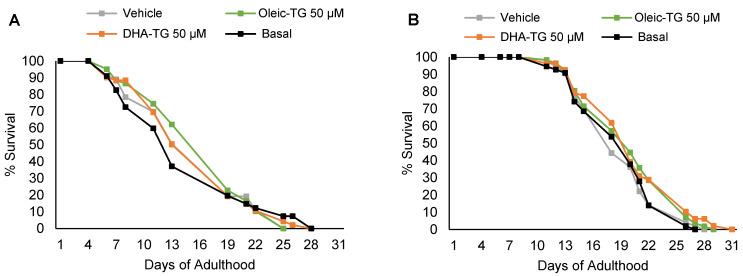
Effects of vehicle, Oleic-TG, and DHA-TG on the lifespan of N2 *C. elegans* at 20 °C. The graphic on the left (**A**) shows the lifespan without FUdR. On the right (**B**), it is represented the lifespan with the use of FUdR.

**Figure 3 cells-12-01932-f003:**
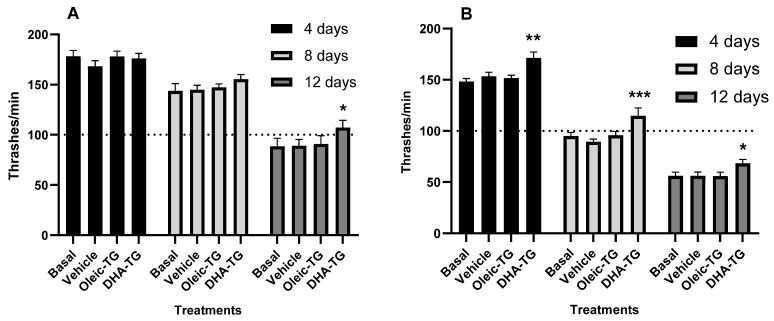
Motility improvements in WT nematodes treated with DHA-TG. The locomotive function was assessed without FUdR (**A**) and with FUdR (**B**). In three independent repeats, a total of at least 30 nematodes per condition and age were analyzed. Data are represented as mean ± SEM, and statistical differences compared to the vehicle were considered significant at *p* < 0.05 (*), *p* < 0.01 (**), and *p* < 0.005 (***) after two-way ANOVA followed by Dunnett’s post-test.

**Figure 4 cells-12-01932-f004:**
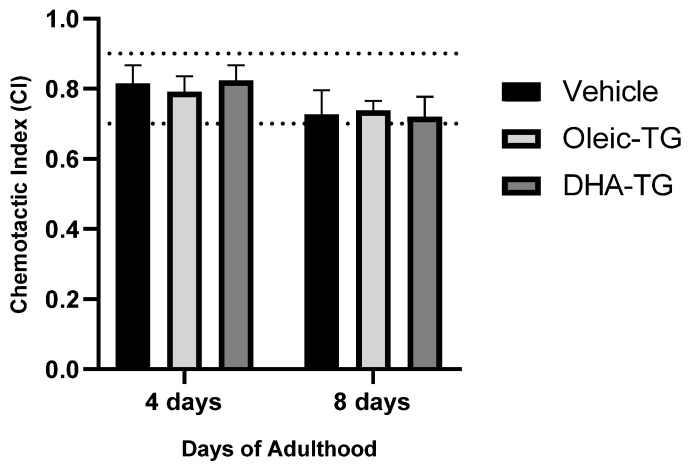
Sensorial response in WT nematodes treated with DHA-TG. Chemotaxis Index (CI) determined in three independent repeats, a total of at least 50 nematodes per condition and age were analyzed. Data are represented as mean ± SEM.

**Figure 5 cells-12-01932-f005:**
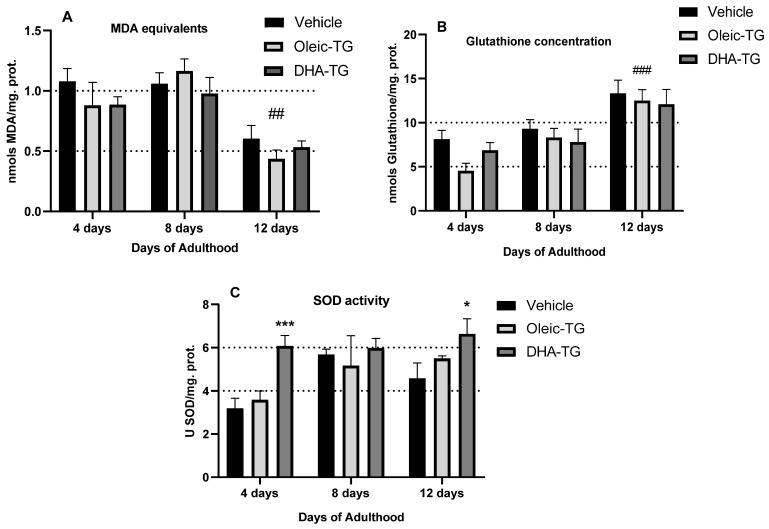
Lipid peroxidation (**A**), glutathione quantification (**B**), and SOD activity (**C**) after Oleic and DHA-TG treatment. Results are presented as mean ±SEM. Statistical differences compared to vehicle of each age are expressed with (*), and statistical differences between ages are expressed as (#). Differences were considered significant at *p* < 0.05 (*), *p* < 0.01 (##), and *p* < 0.005 (***) (###) after two-way ANOVA followed by Dunnett’s post-test.

**Figure 6 cells-12-01932-f006:**
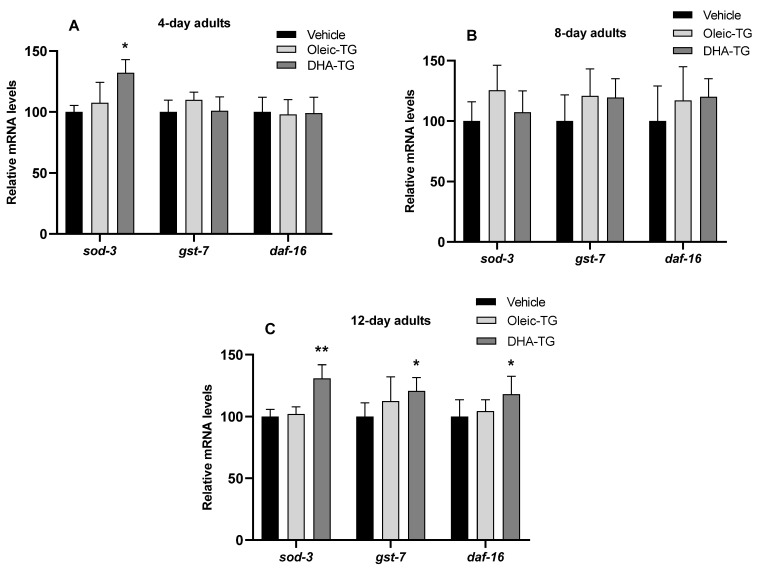
Levels of gene expression of WT nematodes at 4th (**A**), 8th (**B**), and 12th (**C**) days of adulthood after treatment with Oleic and DHA-TG. Relative mRNA levels are expressed as percentages vs. vehicle for each gene and age. Results are presented as mean ±SEM of at least 3 biological replicates in triplicate for each one. Statistical differences compared to the vehicle of each gene are expressed with (*), and differences were considered significant at *p* < 0.05 (*) and *p* < 0.01 (**) after T-Student comparison.

**Figure 7 cells-12-01932-f007:**
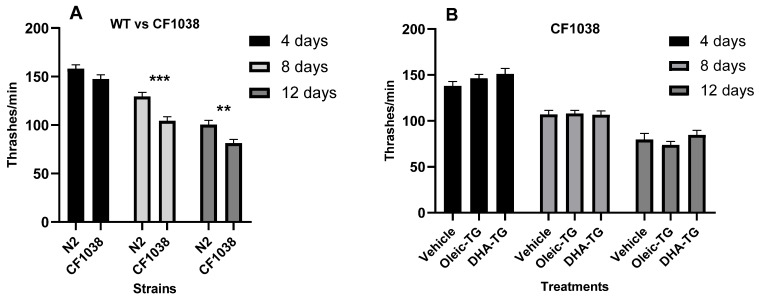
Motility in WT (N2) and DAF-16 deficient (CF1038) nematodes treated with Oleic and DHA-TG. The locomotive function of N2 vs. CF1038 in basal conditions was assessed (**A**), and the effect of DHA-TG in CF1038 was monitored (**B**). In three independent repeats, a total of at least 30 nematodes per condition and age were analyzed. Data are represented as mean ± SEM, and statistical differences were considered significant at *p* < 0.01 (**), and *p* < 0.005 (***) after two-way ANOVA followed by Dunnett’s post-test.

**Table 1 cells-12-01932-t001:** Mean lifespan after Oleic-TG and DHA-TG treatment in N2 nematodes.

FUdR	Treatment	Mean Lifespan	SEM	*p*-Value vs. Basal
No	Basal	14.47	0.89	-
	Vehicle	15.49	0.86	0.636
	Oleic-TG	16.58	0.75	0.262
	DHA-TG	15.78	0.76	0.524
Yes	Basal	18.89	0.60	-
	Vehicle	18.78	0.60	0.919
	Oleic-TG	20.02	0.66	0.126
	DHA-TG	20.33	0.70	0.067

**Table 2 cells-12-01932-t002:** Expression of *sod-3* and *gst-7* after DHA-TG treatment in CF1038 nematodes.

Data	Age (Days)	*mu86* *sod-3*	*mu86* DHA-TG *sod-3*	WT *sod-3*	*mu86* *gst-7*	*mu86* DHA-TG *gst-7*	WT *gst-7*
Mean	4	1.0	0.9	36.4 ***	1.0	1.1	3.9 ***
	8	1.0	1.2	336.0 ***	1.0	1.2	10.4 ***
	12	1.0	1.0	1342.8 ***	1.0	0.8	6.4 ***
SEM	4	0.2	0.1	5.3	0.3	0.2	0.6
	8	0.2	0.1	53.5	0.2	0.5	2.2
	12	0.1	0.2	50.4	0.2	0.1	2.5

CF1038 strain, mutant *mu86* (defective for *daf-16*); N2 strain (WT); *** = *p* < 0.005 after T-Student comparison.

## Data Availability

Not applicable.

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
