# Peer review of "Structured Docosahexaenoic Acid (DHA) Enhances Motility and Promotes the Antioxidant Capacity of Aged C. elegans"

_cells, 2023, doi:10.3390/cells12151932_

Round 1

Reviewer 1 Report

In this study, Mora et al., investigate the impact of a structured triglyceride form of DHA (DHA-TG) on aged populations of C. elegans. They performed a variety of physiological assays (including lifespan, thrashing and chemotaxis) and molecular/biochemical assays to elucidate the response elicited by drug treatment. The treatment yielded a modest effect on some of these assays, and some effects were only detectable in the presence of the drug FUdR. The authors further show that animals which lack functional daf-16 lose any benefits afforded by DHA-TG drug treatment. With major revisions I feel this paper could be suitable for publication.

Line 85 – I’m not sure what “and synapse” is doing in this sentence, and also the usage of reference 25 which has nothing really to do neurotransmission or synapses.

Line 120 – Please be more consistent with the correct nomenclature. In C. elegans reports, the gene and allele are always italicized, ie. daf-16(mu86). It is also incorrect to mention “transgenic” daf-16(mu86). They are not carrying any extrachromosal or integrated arrays. Change to “mutant”.

Lines 147-148 – This concentration of Nile Red seems extremely high! Could this be a typo instead for 1 ug/mL?

Line 167 – “We analyzed at least 30 animals per condition and blindfolded”, change to ‘blinded’.

Lines 173-178 – Chemotaxis assay. I am very confused by how the authors did this experiment. Shouldn’t you put the drugs into the plate first, let it dry, then add the worms to the test plate?

Lines 212-213 – Remove “The superoxide converts a colorless substrate in the detection reagent 212 into a yellow-colored product. Less yellow colour means more SOD in the sample”, or just the second sentence. If you do want to explain this colormetric assay, you should explain this for all your other assays.

Figure 1 – This figure is disorganized and feels unfinished. Please attempt to arrange the images with more regularity to make it more visually appealing. In Panel A, the worms are of clearly different ages with the top animal being a very fresh young adult (no eggs) and the bottom one seems to have one row of eggs (day 1 adult?). For a better comparison, please present worms with better synchronization. I am a bit confused by the choice of treatment. Shouldn’t you also present worms treated with NR alone and then the NR+ DHA-TG emulsion as opposed to just the emulsion without the dye? It seems obvious that the DHA-TG emulsion treated worms without NR would not fluoresce.

 In Panel C, the worm at the top left looks like it is a larval staged worm wheras the others are adults. I suggest reviewing some of the recent worm aging literature where people show ‘stacks’ of worms (popularized by Andy Dillin), where all the worms are facing in the same orientation and are in neat rows. If you use less liquid when mounting on an agarose pad it is much easier to do this. Panel E, I understand that the fluorescence is low, but if you invert the image it might be easier for the reader to see something.

Line 290-291: “shows a tendency (p-value of almost 0.05)” I understand that the authors are saying that the result is “almost significant” but this seems almost unfair to the data. Please just say that it was in the right trend but not significant.

Figure 2: The thrashing numbers reported here are much higher than other studies (ex. doi:10.1242/dmm.036137), can you comment on this discrepancy?

Line 346-349: I do not understand this sentence, also was this experiment (Figure 4) done with or without FUdR?

Line 369-371: Rewrite this sentence and use “first” instead of “in first place” etc. Please read out this study (10.18632/aging.100275) where they actually did calculate lipid peroxidation with age and found a negative correlation.

Line 398: change “was more in depth characterized” to “further characterized” or “further investigated”

Figure 6 – Check nomenclature. The nomenclature here is typical mammalian notation, not worm notation. Please convert genes to all lower case and italicized.

Line 401: “The levels of sod-3 are overexpressed”, change to “the level of sod-3 were elevated”

Line 422: “The factor DAF-16/FOXO is altered” – This is too vague and not true. Are protein levels increased? Is it more DAF-16 translocating to the nucleus? Please clarify.

Figure S1: This is not that helpful but I see what the authors were trying to show. I think this would be better replaced with an end-point PCR gel – ie. just regular RT-PCR (after 35+ cycles) to show that there is no daf-16 band in the mu86 mutant. Since the authors probably have the excess  cDNA stored at -20C it’s just a simple PCR and gel and would make a much nicer figure.

Discussion

Line 465: Is this truly increasing capacity? I thought that even in some of the short-lived animals that SOD enzymes can be upregulated, and that it’s not necessarily that capacity is increased, but that they are stressed and it’s just a generic stress response. Is there a way to know if DHA-TG is not just stressing the worms out?

Overall: I feel that if the above changes are implemented, and perhaps some additional easy behavioural experiments to further nail the ‘neurotransmission’ aspect, I could see this study as being interesting to the C. elegans aging community.  I would like to see some of the levamisole/aldicarb assays like in doi:10.1242/jeb.068734, and also would like the authors to explicitly mention in every experiment in this paper if FUdR was used, since it apparently has such a strong impact in some of the behavioural assays.  

Numerous typos (ex. Line 49 – sever instead of severe, Line 165 – trashes instead of thrashes), mild grammatical errors throughout the manuscript – needs proofreading. Please use the term ‘literature’ instead of ‘bibliography’. Unnecessary usage of the word ‘the’ in front of Table/Figure references in the text.  

Missing abbreviation – the first time you use an acronym like ROS you need to define it

Inconsistent usage of ω-3 LCPUFA with dashes (ex. LC-PUFA).

Reviewer 2 Report

Manuscript Number: cells-2505969

Structured Docosahexaenoic Acid (DHA) enhances healthspan and promotes antioxidant metabolism of aged C. elegans by Mora et al.

In this manuscript, the authors described the  benificial antioxidant effect of DHA-TG on worm motility by DAF-16. It is an intresting work, but the manuscript quality needs to be much improved before it could be accepted for publication in this journal.

Detailed comments and suggestions:

1. Title: antioxdant metabolism should be rewritten as antioxidant capacity, and DAF-16 should be added in the title on a proper form.

2. Abtsract: healthspan can be referred after more healthspan indice have been investigated besides just only a motility index, such as reproduction, antioxidant resistance capacity etc. So, here the authors should narrowed it as locomotivity.

3. Introduction: page3 line88,in contrast animals, the word "animals" was used uncorrect,because C. elegans also is a kind of animal.

line91,in comparsion with mammals were superfluous, it should be deleted.

4. line104,DHA-TG should be first described in details, the readers are difficult to find the difference between DHA and DHA-TG.

5. There are lots of style error, oC is in different style in different sentences; the subscript should be corrected in Na2HPO4 and KH2PO4; second or abbreviated as "S" or "sec", minute to min, hour to h should be consistent, but not be used supersede; mg. should be mg in line225.

6. As the authors addressed in abstract, structured forms of EPA and DHA which have been developed to improve its bioavailability and bioactivity in comparison with conventionalω-3 supplements, then why was not a kind of structured forms of EPA and DHA be used in present work but only DHA-TG was chosen for the study? The impotance should be clearly addressed.

7. The experiment design: DHA-TG was used in the study, and Oleic-TG was used as lipid addition control, but DHA should also be used as another control in present work.

8. Figure1, figure composition is relex, it should be redone.

9.Statistical analysis: the results should be presented as mean ± SD but not mean± SEM.

10. Line270, In the Figure1A-B there should have a comma here.

11. Talbe1 legend, Oleic should be Oleic-TG?

12. Gramma errors in Line299 and line306, consist should be consists; the negative sentence should be used correctly in line346-348,line380-381, line441-442, the meaning is reversed; rewrite line490-491 "the effect of DHA-TG was accompanied neither by a significant reduction nor an increasing of lipid peroxides".

13. Sod3, Gst7, Daf16 should be written correctly in Figur6, Table2 legend, and N2 should be wt, CF1038 should be presented with its gene type.

14. Figure6,only the expression of daf-16 was upregulated is not enough, the activation of DAF-16 should also be investigated.

15. Table2, N2 DHA-TG should be added in this table. It is should be checked that 1342.8 for sod-3 of N2, but in Figure6, it is only nearly 125 or less?

16. Discussion: line507, DHA is different from DHA-TG, the authors should cited references carefully, it possible to induce the readers to  mix the information of DHA and DHA-TG. 

 Gramma errors in Line299 and line306, consist should be consists; the negative sentence should be used correctly in line346-348,line380-381, line441-442, the meaning is reversed; rewrite line490-491 "the effect of DHA-TG was accompanied neither by a significant reduction nor an increasing of lipid peroxides".

Round 2

Reviewer 1 Report

The authors have addressed the majority of my concerns, except that all figures with images of worms still need scale bars. 

There's still some minor grammatical errors and awkward choices of words throughout the manuscript. Please consider a native English speaker to comb through the paper in fine detail, as the transitions and word choices that have been used can be somewhat jarring. 

Author Response

First, we want to thank the reviewer for his/her comments and suggestions.

We added a scale bar to all the pictures, including the pictures in the supplementary material.

Regarding the grammatical errors, our lab mate from the UK reviewed the manuscript and helped us to improve the transitions and word choices. Certainly, our manuscript had awkward choices of words, but we think we changed them for more appealing grammatical options and our manuscript is improved.

Reviewer 2 Report

Manuscript ID: cells-2505969

Although the autors have improved the manuscript, there are still some drawbacks that must be changed before the manuscript can be accepted for publication in this Journal. 

Line377, there are two comma after the word "levels";

Line379, a comma should be used after the reference [58], and rewrite "as happen with MDA levels, there are not many precedents assessing GSH in aged C. elegans";

Figure6A,B and C, sod3, gst7, daf16 should be written as sod-3, gst-7, daf-16, and delete genes. 

None.

Author Response

First, we want to thank the reviewer for his/her corrections.

We made all the changes proposed by the reviewer and we changed some paragraphs of the manuscript to make it more appealing.